# Highly Concurrent TCP Session Connection Management System on FPGA Chip

**DOI:** 10.3390/mi14020385

**Published:** 2023-02-03

**Authors:** Ke Wang, Yunfei Guo, Zhichuan Guo

**Affiliations:** 1National Network New Media Engineering Research Center, Institute of Acoustics, Chinese Academy of Sciences, No. 21, North Fourth Ring Road, Haidian District, Beijing 100190, China; 2School of Electronic, Electrical and Communication Engineering, University of Chinese Academy of Sciences, No. 19(A), Yuquan Road, Shijingshan District, Beijing 100049, China

**Keywords:** TOE, FPGA, session management, multi connections

## Abstract

Transmission Control Protocol (TCP) is a connection-oriented data transmission protocol, and it is also the main communication protocol used for end-to-end data transmission in the current Internet. At present, the mainstream TCP protocol processing service is implemented by software running on the Central Processing Unit (CPU). However, with the rapid growth of transmission bandwidth and the number of connections, the software-based processing method is not ideal in terms of delay and throughput, and also affects the processing performance of the CPU in other applications such as virtualization services. Moreover, other hardware solutions can only support a limited number of TCP session connections. In order to improve the processing efficiency of the TCP protocol and achieve highly concurrent network services, this paper proposes a TCP offload engine (TOE) prototype system based on field programmable gate array (FPGA) chips. It not only provides hardware-based data path processing, but also realizes hardware management of large-scale TCP session connection status through a multi-level cache management mechanism. Studies have shown that this solution can support 100 Gbps high-performance throughput characteristics, and allow concurrent processing of hundreds to 250,000 TCP connection state hardware maintenance on a single network node, improving the overall performance of the network system.

## 1. Introduction

In the contemporary Internet, with the rapid development of the traffic carried by by network, the transmission bandwidth and the number of network connections have ushered in an explosive growth, which makes the data path become wider and more dense. The frequent interaction of massive data and the continuous abundance of Internet services have put forward higher requirements on the transmission performance and flexibility of network equipment, and the network architecture has become more complex. The introduction of Network Functional Virtualization(NFV) technology allows the flexible definition of the network in the form of software on general server equipment, thus bringing huge benefits to the management and maintenance of the network [1,2,3]. By migrating complex network functions to run in virtual machines or containers on servers, it is also possible to provide more diverse network services on a single computing node. This is also conducive to efficient utilization of network equipment resources and bandwidth. However, limited by Moore’s Law, general-purpose CPU performance growth slows [4], which makes it difficult for network communication services that rely entirely on software design to be suitable for high-performance application scenarios. Especially in virtualized cloud data centers, highly concurrent network data brings huge network protocol processing overhead, which largely limits the available CPU cycles of virtualized network applications, thereby affecting the quantity and quality of network services. In a 10 Gbps network, the packet protocol processing overhead will require half of the computing power of an 8-core high-end CPU [5]. Therefore, how to improve the processing efficiency of the network protocol has become an important bottleneck in improving the performance of the network system.

As the core component of the current network communication protocol, TCP provides reliable end-to-end data transmission, and has been widely used on computing nodes in the end, edge, and cloud. But it is also the most complex and demanding communication protocol, so the resource overhead it brings cannot be ignored. This is because of its connection-oriented transmission characteristics. In the process of data transmission, it is necessary to provide connection state management overhead for each session connection. Specifically, it can be divided into many operations such as reading, judging, updating, and saving the connection state. Every time a TCP session connection is added, its maintenance cost will also increase accordingly. The second is about the order-guaranteed transmission of data. For the TCP protocol, the sender and receiver need to make judgments based on the accumulation of sequence numbers to ensure that the data packets can be delivered in the original order. This will also introduce a large number of small packet interruption transmission problems. In addition, it also includes various optimization strategies for reliable data transmission such as flow control and congestion control. The traditional TCP protocol processing is realized through the operating system kernel installed in the general-purpose processor chip, and it can support a complete TCP processing mechanism. However, the versatility of the kernel protocol stack also brings complicated branch judgment and data copy problems, resulting in a large amount of CPU overhead. Therefore, it is not suitable for increasingly higher network transmission bandwidth and large-scale network session connection management.

In order to improve network transmission performance, researchers have proposed many optimization measures, which include improving the existing TCP/IP kernel protocol stack to ensure good compatibility with general-purpose operating systems [6,7,8].The second type of work is to use the user-mode protocol stack that completely bypasses the kernel [9,10,11,12], which can reduce the overhead of kernel data replication and interrupt processing. However, no matter the improved strategy proposed in the kernel or in the user space, it still requires high software data distribution cost and state maintenance overhead. For example, the fast path method proposed in Reference [12] is used to reduce TCP processing overhead and provides processing efficiency up to 40 Gbit/s line rate. However, for communication-intensive applications, this solution needs to spend up to 74% of CPU cycles on network packet processing.

The heterogeneous model using TOE-assisted CPU has become a new solution. TOE can offload part or all of the TCP/IP protocol stack to Application Specific Integrated Circuit (ASIC), FPGA and other hardware devices for implementation. In this way, the host CPU processing overhead brought by the communication primitives of the operating system is effectively reduced. Among them, ASIC [13], as a special-purpose processing chip, provides customized processing for network functions and has high transmission performance, but due to the limitation of its design structure, there are bottlenecks in performance and scalability. In contrast, the FPGA Smart Network Interface Card (SmartNIC) [14,15] as a programmable hardware acceleration unit, not only has high performance at the hardware level, but also allows flexible adjustment of design functions through rewiring or parameter configuration. It is more suitable for this kind of virtualized cloud data center network that has high requirements for throughput and scalability. At present, most commercially successful TCP offload functions are stateless offloads for traffic-intensive applications. Stateless means that the internal storage of the Network Interface Card (NIC) is read-only during packet processing. For example, TCP Checksum offload [16], TCP segmentation offload (TSO), etc. The core function of the TCP transport protocol is to maintain the connection state, which not only has the characteristics of data-intensive services, but also involves a large number of control logic implementations. At present, there are few related studies in this area, and some commercial accelerator cards [17,18] that support stateful TCP function offloading are mainly aimed at high-frequency trading scenarios with ultra-low latency requirements. To minimize latency, these schemes only support a small number of session connections. This solution is mainly aimed at the high throughput and large-scale network session connection management requirements of the virtualized cloud data center network, and proposes a prototype of a high-concurrency TCP session connection management deployed on the FPGA SmartNIC. The main contributions of this paper are as follows:(1)We implemented the hardware processing logic of the TCP protocol in the FPGA, which includes protocol parsering and frame encapsulation of the sending and receiving data paths, as well as connection state maintenance including establishment and teardown.(2)The solution provides a multi-level state management mechanism, using on-chip storage resources and off-chip storage devices to jointly maintain the TCP session connection state including four-tuples. The method supports high-performance data transmission for 128 TCP connections and state maintenance for hundreds of thousands of long TCP connections. Thereby achieving a balance of high throughput and high scalability within limited hardware resources.(3)The solution is completely designed with the hardware description language Verilog, and provides stateful data processing logic based on hardware timing transmission, which has high stability and portability.

The rest of this paper is organized as follows. The Section 2 reviews some existing TOE schemes. Section 3 introduces the hardware offloading implementation of the high concurrent TCP session connection management mechanism based on the FPGA SmartNIC, and elaborates on the implementation details of data sending and receiving and state management. In Section 4, we deploy the TOE on a Xilinx FPGA platform and evaluate the data transfer performance of this method. Finally, we summarize the scheme.

## 2. Related Work

Over the past few years, there have been several case studies on TOE in academia and industry. Reference [16] deployed a TCP checksum offload scheme for 100 Gbps high-speed networks on a programmable acceleration platform. The hardware-based low-latency checksum processing operation is realized through the pipeline design, which alleviates the data path processing overhead. Reference [19] offloads the data path of the TCP protocol to the network processing unit (NPU), and adopts a fully customizable processing method to support the high throughput and high flexibility requirements of the data center network. But it mainly targets the TCP data path of the established connection to avoid constructing complex control logic in the NIC. Reference [20] present a dual-stack TCP design that splits functionality between the host and NIC stacks. The host stack holds the master control of all TCP operations, and offloads connection establishment and teardown to the NIC, which avoids the frequent interaction of small control packets to a certain extent. However, state maintenance for established connections remains implemented in the host, and thus still involves Direct Memory Access (DMA) transfers of a large number of acknowledgment packets with smaller sizes, and software handling of order-preserving transfers for each stream. Reference [21] achieved 10 Gbps throughput performance on a single TCP connection, providing end-to-end accelerated processing services for short-distance or low-latency applications in tightly coupled networks. Reference [22] also proposed a hardware TOE for 10 Gbps Ethernet. It supports the establishment and closure of TCP sessions, as well as various optimization options including timestamps. But it can only achieve 10 Gbps throughput performance when the size of the transmitted data packet is a jumbo frame with a length of 65,536 Bytes. The multi-session TOE for delay-sensitive applications proposed in Reference [23] allows low-latency and scalable TCP session management through kernel bypass technology and hardware-based parallel connection management. But this solution needs to create an instantiation unit for each connection, and each instance implements a complete processing pipeline. Therefore the number of supported connections is limited and resource utilization is inefficient. Reference [24] implemented a video-on-demand-oriented asymmetric TCP/IP offloading in hardware. As a client, it provided a high throughput performance of 40 Gbit/s and connection management of more than 10k channels. However, as a server, it also only supports a limited number of connections and low transmission efficiency. Reference [25] and Reference [26] used high-level synthesis (HLS) to implement scalable TOE of 10 Gbps and 100 Gbps respectively, supporting large-scale session connection management. However, they need to consume a lot of expensive on-chip memory resources for maintaining connection state. To further reduce BRAM overhead, Reference [26] uses the three-tuple to obtain the sessionID, which is not suitable for use in virtualized networks with multiple local (Internet Protocol) IP addresses. In addition, as a high-level hardware programmable language, HLS needs to go through complicated translation processing when used, which also increases the design instability for complex protocol designs such as TCP. Reference [27] implements TCP services using extended Finite State Machines (XFSMs). The platform-independent programming abstraction interface it provides allows flexible deployment on a variety of platforms, but specific details about throughput performance and scalability are not disclosed here. Reference [28] provides a collective communication library for FPGA-to-FPGA network for Xilinx devices, and realizes session management of 1000 connections and 100 Gbps transmission performance in Xilinx Vitis. In addition, there are currently many commercial TOE solutions for high-speed networks [17,18,29,30], and they are mainly aimed at high-frequency trading scenarios with high latency requirements. The number of supported connections is limited, and it is often provided in the form of a netlist file, which has poor scalability.

By analyzing the TOE schemes mentioned above, we found that the current mainstream TOE schemes cannot well support the hardware management of large-scale TCP session connection states. Therefore, we propose a hardware model for high-speed network-oriented large-scale TCP session connection management, which provides a new solution for high-concurrency session connection management in virtualized cloud data centers.

## 3. TOE Architecture

The hardware model of large-scale TCP session connection management implemented in this solution is shown in Figure 1, which is mainly composed of two parts: data path processing logic and shared connection state management logic based on multi-level cache units. Among them, the sending and receiving data path provide the parser and encapsulation operation of TCP packets, which includes the processing logic of various TCP packets distinguished according to the TCP flag field, as well as the interface for reading and writing connection status and event triggering. Connection state information is stored and shared between the sending and receiving paths through a variety of shared data structures. Here we use the combination of FPGA on-chip storage unit Block Random Access Memory (BRAM) and off-chip storage unit Double Data Rate Synchronous Dynamic Random Access Memory (DDR). We place the more frequently used connection state in BRAM and store the lower priority connection state in DDR. This method allows to provide 100 Gbps high throughput performance for 128-way connections, and provides state maintenance for large-scale long connections. Next, we introduce the main functional logic involved in the design in detail.

### 3.1. RX Processing Core

The TOE offload engine provided in this study provides a bidirectional data transmission channel. The RX processing core is responsible for receiving and parsing TCP packets, and sending the payload and metadata information to the RX buffer. A series of status checks and update operations are also performed before this.

When the TCP packet containing the pseudo-header arrives at the TOE module, it is first necessary to separate the metadata and valid fields. The TCP metadata separation module provides the extraction logic of the configurable header length, and adjusts it to the little-endian mode for easy parsing and then outputs it. Extract the destination port and four-tuple {SIP, DIP, Sport, Dport} information in the header to initiate a request to the port management table and the connection state management table respectively. Afterwards, the result and the meta signal of the data packet are synchronously sent to the RX processing logic for further inspection processing. The specific inspection process is shown in Figure 2. First we need check whether the destination port of the data packet is monitored. If the port is in the listening state, it is further judged whether the connection is valid, that is, whether the currently stored connection socket information is consistent with the packet four-tuple. The receiving state machine we provide makes specific distinctions for TOE packets according to the flag. For an active session connection, if a RST packet is received, we need to close the connection. When a packet marked SYN is received and the current connection state is CLOSED, we update the connection state to the SYNC_RCVD state. Then the doorbell event management module will notify the TX logic to generate a synchronous confirmation packet for the second handshake. For packets such as SYN_ACK, FIN, ACK, etc., state judgment and serial number inspection operations are also provided respectively. When the check is completed, we will update the connection’s RX control state and connection state based on the processing result. If the packet has a payload, we will set the extraction period according to the length of the payload, and send the payload data to the receiving buffer in turn to wait for the upper-layer application to read. For packets that do not meet the requirements, they will be discarded through the drop logic. The RX processing core can not only receive data requests from the server, but also receive confirmation responses from the client. If there are unacknowledged packets in the sending window, but the receiving end receives three consecutive identical acknowledgment signals, the fast retransmission logic will be triggered to ensure the reliability of data transmission.

### 3.2. TX Processing Core

The TX processing core provides the data path opposite to the RX logic and is responsible for merging and forwarding packets. The generation of each sent packet is triggered by a doorbell event. These doorbell events include confirmation reply events from the RX processing core, passive handshake and disconnection events as the server, active establishment and release connection events from upper-layer applications, and packet sending events. After the application logic receives the sending request, it also needs to obtain the connection status according to the four-tuple, and send the query result to the TX processing core. Unlike the receive path, in order to ensure reliable transmission of data, we provide each connection with 64 KBytes size of the off-chip buffer, to enable the update and maintenance of the send window. For the packet of the connection establishment request, the TX processing core generates a random sequence number through the four-tuple hash and nanosecond time stamp information, which is used for the sequential transmission of the packet. Finally, we pack the generated packet header information and payload field into a new data packet, and then send it out in the form of Advanced eXtensible Interface Stream(AXIS) bus. Then update the connection control information including the sequence number.

### 3.3. Multi-Level Cache State Management Mechanism

As a key feature of this solution, the multi-level cache state management mechanism is used to realize the high throughput performance of 128-way connections and maintain the state of large-scale long-session connections.

**L2 Table:** L2 Table provides a variety of control information for connection state management, which can be shared in the sending and receiving path. As shown in Figure 3, it consists of an Exact Match (EM) table and four sets of connection state control tables (Socket Table, Rx_Ctrl Table, Tx_Ctrl Table, APP_Ctrl Table). The EM table implemented based on the register group provides accurate matching of the four-tuple information, and obtains the matching hit identifier and index information according to the input four-tuple key value. The Socket Table stores the status information of the current connection and the hash value of the four-tuple. The other three groups of entries respectively store the connection state information written by each processing logic control. Careful partitioning of the data structure for storing connection states can help reduce the number of operations and improve retrieval efficiency. For example, in the packet receiving link, the RX processing core is responsible for updating the local and peer acknowledgment numbers, while the update of the local sequence number is implemented by the packet sending logic. In addition, in the data forwarding link, the APP logic updates the write pointer, the TX logic updates the read pointer, and the RX logic updates the confirmation pointer after receiving the confirmation signal from the peer, thereby dynamically estimating the size of the sliding window during the forwarding process. After receiving the four-tuple key, we first send it to the EM table to check whether it is hit. If the target connection exists, then further obtain the specific connection status in each control information entry according to the index value. The whole process can be completed within two clock cycles. After the data processing is completed, we update further the corresponding connection status. In order to improve the retrieval efficiency of table items, we use registers to design EM tables. As the number of management connections doubles, the Look-Up-Table (LUT) resources they occupy grow exponentially. After analysis, it is a reasonable choice for the currently selected FPGA platform to provide high-frequency access to the 128-way connection status. Of course, we also provide parametric design, which allows flexible adjustment of the depth of the table according to different FPGA models.

**L3 Table:** Although the L2 Table provides efficient retrieval of the connection state, it is limited by the resource and wiring complexity and supports a limited number of connections. In order to maintain the state of larger-scale long-session connections, we introduce a three-level cache table on this basis. This is a design method based on DDR. Compared with expensive on-chip storage resources, DDR has larger storage space and lower cost, but the read and write efficiency is relatively low. Here, we allocate 64 Bytes of space for each connection to store the connection state, including a certain amount of redundant space to facilitate future function expansion. A piece of DDR space of 256 KBytes is enough to support the storage of hundreds of thousands of connection states, so as to realize the large-scale session connection state management. For L3 Table access, we use a lookup method based on hash values. When the L2 Table misses, the packet flows to the slow path for L3 Table lookup, and the query result is updated to the L2 Table. The specific update logic will be introduced in detail in the following chapters.

**L1 Table:** The branch judgment of the fast path and the slow path introduces a certain processing delay in the transmission process of the data packet. Although there is a certain loose coupling phenomenon between the sending and receiving logic in the state sharing of the TCP connection, for the same connection, the sequence number update has a strict order dependency. Excessive latency affects transfer performance on a single connection. In order to reduce the delay overhead in the state access process, this solution introduces a first-level cache table inside each processing core to store the connection state update of the current processing logic. For example, in the data sending link, every time a data packet is generated, the TX processing core updates the sequence number to the L2 Table and the L1 Table at the same time. When updating the L1 Table, it is also necessary to add timestamp information to the updated information. When data needs to be sent again on the connection, it is first necessary to obtain the timestamp information in the L1 Table for judgment, we need to read the status information in the L1 Table to make a judgment. According to statistics, in each path, the maximum budget window from reading the connection state to updating is 160 ns. But the introduction of L1 Table can reduce the minimum frame interval to 20 ns. This approach allows high throughput performance close to line rate on a single session connection.

### 3.4. State Table Update Logic

Figure 4 shows the specific logic of obtaining the connection status of the receiving path. First, the L2 Table is accessed. If it is hit, the query result is sent to the RX processing core through the fast path. After the processing is completed, the L1 Table and L2 Table are updated respectively. If it is not hit, it is sent to the slow path to query the L3 Table. Here we obtain the four-tuple information of the packet from the packet header for hash calculation, and initiate a search command to the third-level buffer stored in DDR according to the hash value. Regardless of whether the query result is valid or not, it is sent to the subsequent processing logic for judgment. And the cold connection in the L2 Table is replaced. The so-called cold connection refers to the connection with the lowest access frequency in the secondary table at present. Here we use the hot management module (Hot mgr) to maintain the heat of the connection status stored in each entry. Every time we initiate a query request to the L2 Table, we will adjust the entry where the hit connection is located to the highest heat. Therefore, we can also judge the access frequency of the connection through the change of the heat of the connection. When new connection data arrives, the new connection information can be replaced by the connection state with the lowest popularity through the table entry replacement unit (Table Replace). If the cold connection is not closed, the connection status needs to be written back to the L3 Table, waiting for the next access. This method can realize dynamic access to large-scale long-session connections. This method is also applicable to the data sending path.

## 4. Performance Evaluation

We have discussed in detail about scalability and flexibility in the previous chapters. In this section we will conduct experiments to evaluate the throughput performance of this scheme design. We deploy TOE on an FPGA SmartNIC model XILINX Zynq UltraScale+ ZU19EG, as shown in Figure 5. The SmartNIC is installed on a Dell R740 server. In addition to the MIG IP core used to control the DDR4 peripheral pins, the core function modules of this program are completely designed with Verilog hardware description language, and the working clock is 250 MHz. In performance evaluation, we generate test data through a TCP packet generation logic (pkt_gen) deployed on FPGA. And according to the connection mode shown in Figure 6, the loopback path formed by two TOE models was used for performance test. Because the RX path does not require the off-chip buffer to store data, it has higher transmission efficiency. Therefore, we first test the data receiving performance of TOE by bypassing the buffer. Then the DDR data buffer is connected to test the data forwarding performance of the TOE system.

The specific test process is as follows: First, pkt_gen sends a connection establishment request to TOE0. TOE0 generates a random sequence number for the connection request and constructs a SYN packet including a pesudo header. Then set the corresponding connection state to SYN_SENT according to the four-tuple information. After TOE1 receives the SYN packet, it first checks whether the target port is in the listening state. After that, the state storage space is also allocated for the new connection, and the connection state is updated to SYN_RECD. Then reply the second handshake signal SYN_ACK to TOE0. After the RX processing core in TOE0 checks the received data packet, it replies with the third handshake data packet. Through the three-way handshake, the connection between the two parties is established. Afterwards, pkt_gen generates test packets to each connection according to the parameterized configuration. The statistical module provides a dynamic record of transmission performance, and can be presented in a graphical interface through the Integrated Logic Analyzer (ILA) provided by the vivado tool.

Figure 7a,b respectively record the transmit and receive performance of the TOE of this solution. It can be seen from the results in the figure that, without the limitation of the optical port module, this solution can provide the transmission performance of receiving 100 Gbps and sending more than 85 Gbps for the connections that completely uses the BRAM storage state. Due to the use of the state management mechanism of the three-level cache, this solution can provide the same transmission performance no matter it is tested on a single connection or 128 connections. For the cold connection test, we completely shield the hit signal of the L2 Table, which makes each lookup operation need to read and write the connection state from the DDR. We poll the test packets on 1000 TCP connections. Since the L2 Table only supports the storage of 128 connection states, it is necessary to write a connection state from BRAM back to DDR for each data transmission, and write the new connection state to BRAM. From the results shown in the figure, it can be seen that, if the connection status is queried from DDR every time, this solution can provide 85 Gbps of RX and 61 Gbps of TX throughput performance for 8192 Bytes packets. This test provides the lower limit of TOE transmission performance. The proposed method also verifies the feasibility of dynamically updating the connection information between BRAM and DDR according to the connection heat. For a 16 G DDR space, 256 KBytes are used to build L3 Table, and the rest space is used to build send buffer, which is enough to support the state storage of 250,000 TCP connections. This approach can even provide tens of millions of levels of hyperscale connection state management without considering the data buffer overhead. The figure also shows the throughput performance of the Linux kernel stack using the FPGA chip as a NIC. We use the iperf tool to test. When the number of iperf threads reached 16, the throughput performance reached the maximum, but this was far inferior to the experimental results of this scheme. And, the solution is completely implemented by hardware, which can perform end-to-end data transfer of multiple session connections without CPU involvement, so the design has a low system cost.

In addition, we further compare the design and the existing TOE solutions that support hardware connection state management. It can be seen from the Table 1, compared with the scheme of other literature, this solution can support more TCP session connections and has better scalability. And our work is also applicable to high bandwidth applications.

In terms of resource utilization, this design provides 8 BRAMs (0.81%), 7952 LUTs (1.52%), and 256 KBytes of DDR space (0.02‰) for the storage of 200K connection states. Compared with the method of completely using expensive BRAM for state storage, this design is a more resource-friendly solution, which provides the possibility for large-scale session connection management.

## 5. Conclusions

In this paper, we introduce a highly concurrent TCP session connection management mechanism deployed on a programmable acceleration device FPGA. The design provides hardware-based TCP session connection establishment and unloading functions, and allows orderly transmission of data packets through sequence number verification. For high concurrent access, the multi-level cache management mechanism provided by this solution allows the states of hundreds to hundreds of thousands of TCP session connections to be maintained simultaneously on a single network node. We use a small amount of on-chip storage resources to achieve efficient access to the state of 128 TCP connections, and provide a throughput performance of 100 Gbps on the receiving side and over 85 Gbps on the sending side. In addition, we can also store a larger-scale TCP connection state in an off-chip storage unit with a large size, and replace the connection information between BRAM and DDR according to the frequency of connection access. This design is beneficial to implement arge-scale, end-to-end data communication in virtualized cloud data centers. Future work includes further optimization of TOE, for example, providing processing logic for timer events, enabling out-of-order packets to be reordered, and hardware solutions to various congestion problems faced in the actual Internet. These features will facilitate highly reliable end-to-end data communication.

## Figures and Tables

**Figure 1 micromachines-14-00385-f001:**
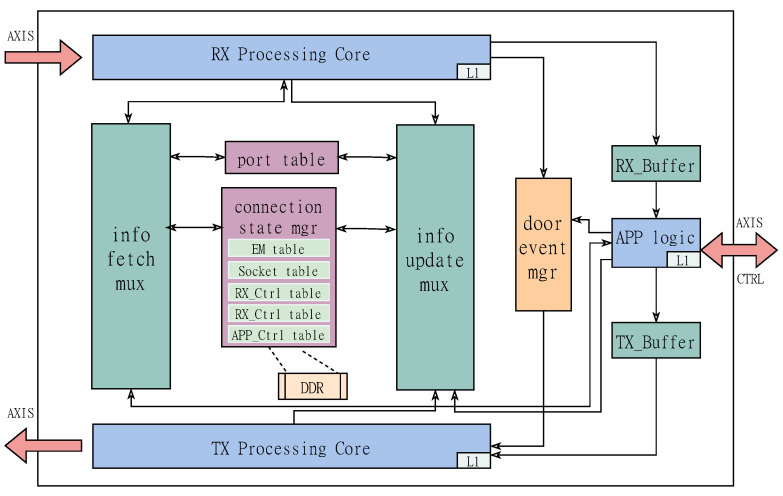
TOE system structure diagram.

**Figure 2 micromachines-14-00385-f002:**
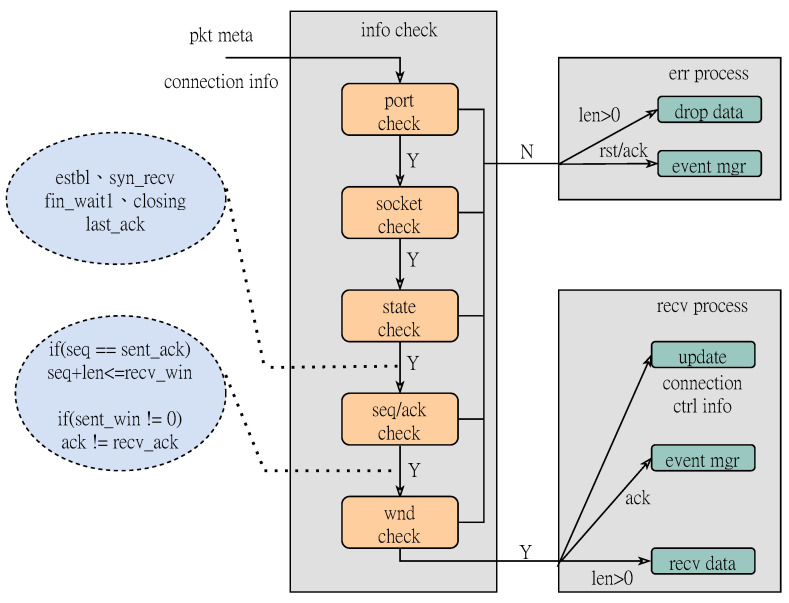
Rx data receiving flowchart.

**Figure 3 micromachines-14-00385-f003:**
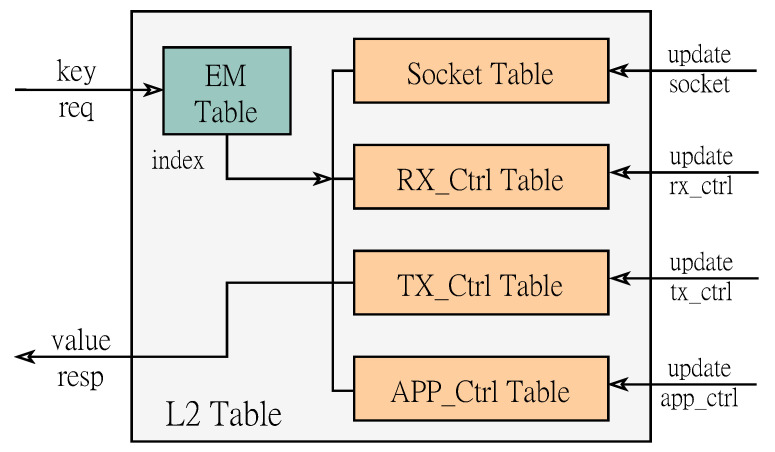
State management structure diagram.

**Figure 4 micromachines-14-00385-f004:**
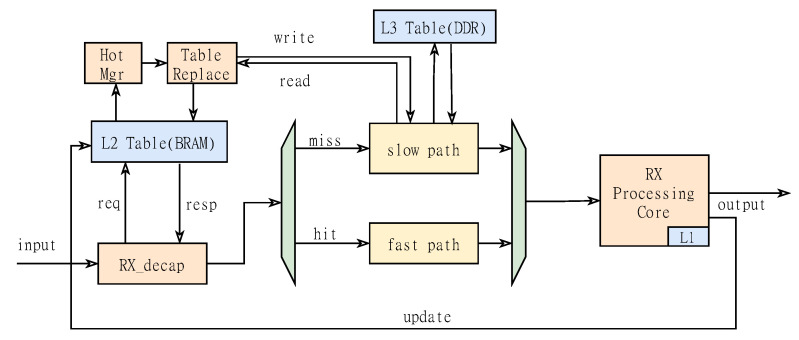
Schematic diagram of state table hot update.

**Figure 5 micromachines-14-00385-f005:**
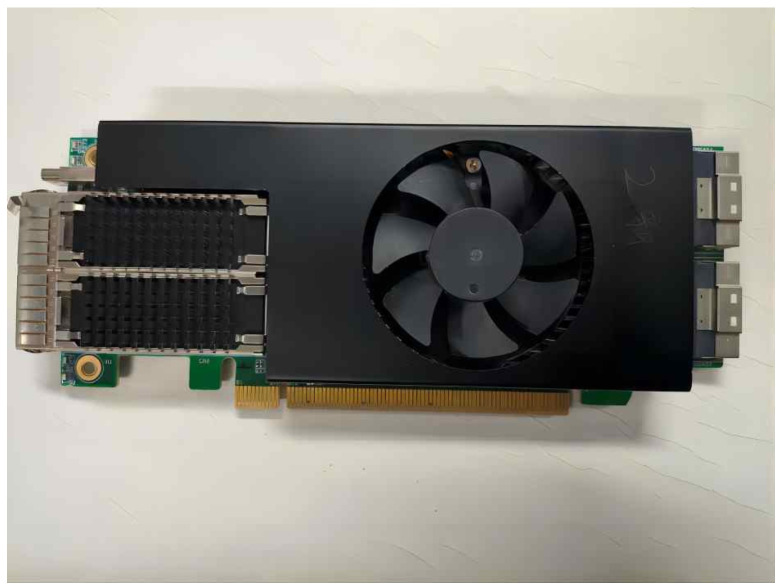
TCP Offload Engine (TOE) on FPGA chip.

**Figure 6 micromachines-14-00385-f006:**
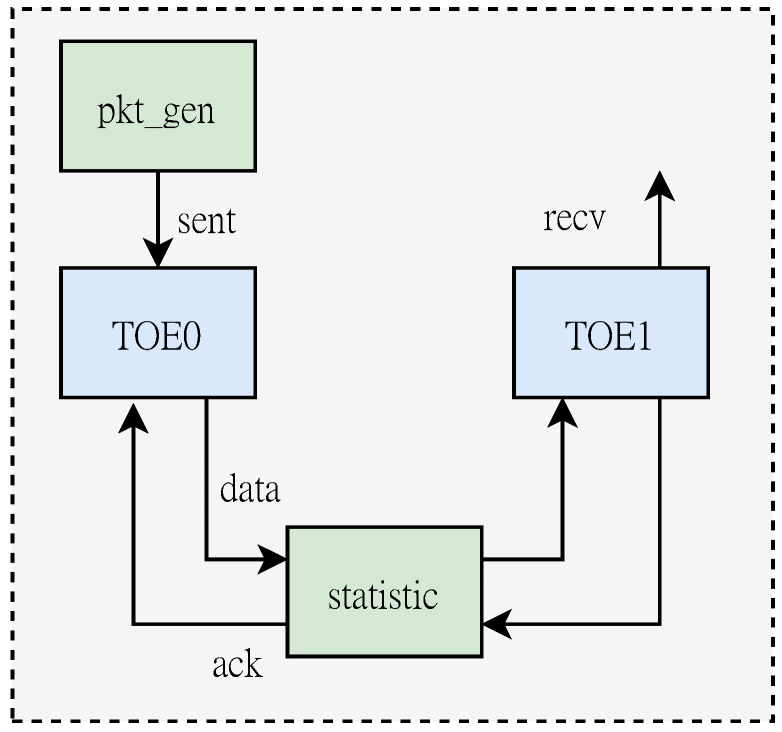
Diagram of test signal transmission.

**Figure 7 micromachines-14-00385-f007:**
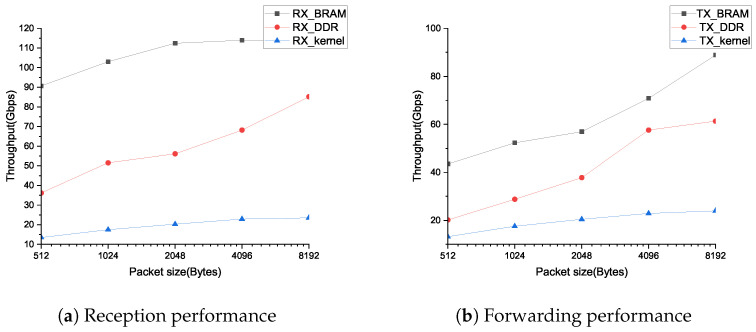
Transmission performance diagram.

**Table 1 micromachines-14-00385-t001:** Comparison of the TOE of this design with previous work.

Design	Number of Connections Supported
Reference [17]	64
Reference [18]	1
Reference [21]	1
Reference [22]	few
Reference [23]	few
Reference [24]	20,480
Reference [25]	thousands
Reference [26]	10,000
Reference [29]	32,000
Reference [30]	1000
Our work	250,000

## Data Availability

All the necessary data are included in the article.

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
