# Peer review of "Highly Concurrent TCP Session Connection Management System on FPGA Chip"

_micromachines, 2023, doi:10.3390/mi14020385_

Round 1

Reviewer 1 Report

The paper is well organized and motivated. The literature review is thorough and the limitations of the state-of-the-art technique are described clearly. The methodology is described comprehensively. The experimental setup is solid. The reviewer has only two comments and some corrections for grammatical errors.

Line 36, can the authors add a reference to the claim - "... will require half of the computing power of an 8-core high-end CPU"?

For the experimental results, can the authors add also the CPU utilization for different number of TCP connections when the TOE is applied?

Grammatical errors:

Line 33, highly concurrent

Line 94, FPGA

Line 109, when TCP offload engine is mentioned in the context, the acronym TOE should be used consistently.

Line 120, Reference, singular.

Line 148, only

Line 174, paths provide

Line 204, TCP

Reviewer 2 Report

The paper introduces a TCP session connection management mechanism deployed on a programmable acceleration device FPGA.
Pros:
The paper is interesting and has useful experiments.
The idea is good and the method used is consistent.
Cons:
There are a few grammatical errors and typos in the paper: (For example ‰)
The performance evaluation section can be improved with a comparative analysis with similar studies or default mechanisms.
 There is no information about future works.
Can this method be applied easily to applications?
